# Increasing the Efficiency of Aircraft Ground Handling—
# A Case Study

Stanislav Szabo [1] , Marek Pilát [1], Sebastián Makó [1,*] , Peter Korba [2], Miroslava Čičváková [1] and Ľubomír Kmec [3]

1   Department of Air Traffic Management, Faculty of Aeronautics, Technical University of Kosice, 040 01 Kosice, Slovakia; stanislav.szabo@tuke.sk (S.S.); marekpilat1990@gmail.com (M.P.); miroslava.cicvakova@tuke.sk (M.Č.)
2   Department of Aeronautical Engineering, Faculty of Aeronautics, Technical University of Kosice, 040 01 Kosice, Slovakia; peter.korba@tuke.sk
3   Faculty of Management, University of Presov, 080 01 Presov, Slovakia; lubomir.kmec@gmail.com
*   Correspondence: sebastian.mako@tuke.sk

**Abstract:** This research was focused on increasing the efficiency of aircraft ground handling at the airport. The main goal of the research was to improve the individual processes that are part of the aircraft ground handling in order to speed up this operation, as well as improve the turnaround time between individual flights to enhance the overall throughput of airport stands. The objective of the research was to measure the times of standard airport processes that are part of the aircraft handling, measure the turnaround time between individual flights at the selected airport and increase the efficiency of each process that was measured. After the measurements had been taken, changes were introduced, and the times were measured again. The changes were mainly focused on the following aspects: the position of ground handling equipment before the arrival of the aircraft, the deployment of staff, and the routes taken by ground handling equipment. The measurements were taken during the summer on a fixed stand, with a standard number of employees and with the same type of aircraft. In total, 78 measurements were taken in which 2340 partial times were measured during the entire course of aircraft ground handling before departure to the next destination. After the changes were implemented, the same measurements were taken again to see if the changes that had been implemented could speed up the overall process of the aircraft ground handling. Subsequently, all data were evaluated using statistical methods. All measurements were done at the Košice airport.

**Keywords:** airport; ground handling; airplane; efficiency





## 1. Introduction

Nowadays, logistics is a dynamically developing discipline. By definition, logistics is an interdisciplinary science that improves the quality of business processes and allows companies to respond to market and customer requirements more quickly [1,2]. The main reason for conducting this research was the need to find new ways to improve quality that meet international standards and reflect the streamlining of services in transport companies. This research was focused on the reduction of the ground handling time and enhancement of the position of aircraft ground handling means [3–5].

An important stage in the process of increasing efficiency is the selection of a suitable criterion. In general, the criterion must meet two basic conditions. It must express the real interest of the airport operator and it must also be well quantifiable—time [6–9]. The aircraft ground handling of one aircraft is the time from on-block to off-block, consisting of several processes where personnel and ground handling equipment are needed [10–13]. The purpose of speeding up the aircraft ground handling is to provide ground handling to as many airlines as possible and to improve the overall throughput of airport stands [14–16]. The time required for the aircraft ground handling is directly dependent on the logistics of the procedural steps which are required [17,18]. A direct factor is the time required

for the handling, which depends on the type of aircraft, the number of personnel, as well as resources required for each task, their appropriate order, and the speed of their management within the limits of safety and technical progress [19]. The indirect factors influencing the length of the aircraft ground handling are the number of disembarking passengers, i.e., the amount of their baggage, the weather affecting safety at work, or the condition of the interior of the aircraft after the arrival [20–22]. Each aspect of aircraft technical clearance consists of procedures which include operations that can be divided into two categories: fixed and improvable [23,24]. One of the objectives of such research is to test a model that would reduce the time required to obtain all improvisable aspects of aircraft control services as much as possible [25,26]. The model consists of measuring the times of individual operations performed during the individual parts of the aircraft ground handling, as well as the total time of the aircraft's stay on the ground between flights [27–30]. The ground control structure is intended to help maintain the required pace. The exchange of information at the operational level in cooperation with decision-making at the airport significantly affects the efficiency of the performance of ground handling tasks. This is accomplished by designing various heuristics for ground handling tasks [31–33]. The aircraft ground handling model is a priority for each airport, directly affecting the quality of service, overall capacity, and financial results, and therefore its efficient operation is necessary [34–36].

Changes in the process of the aircraft ground handling should focus on the positions of ground handling equipment before the arrival of the aircraft, the deployment of staff, routes of ground handling equipment and others [37,38].

## 2. Methodology

The overall approach to solving the above problem, which is either inefficient use of the ground handling equipment or long delivery times of the ground handling equipment, was to monitor the ground handling reference values at the airport to reduce the total time the aircraft stayed on the ground. After increasing the efficiency of aircraft ground handling, the time and individual operations was measured and the overall reduction of the time of the aircraft ground handling procedures was monitored. One of the objectives was to measure the values of the time of the total turn of the aircraft and, subsequently, to measure the partial tasks of aircraft ground handling. The research focused on the means needed for the aircraft ground handling, division of employees for individual tasks, location of means, number of means and others. The aim was to work towards the fastest possible turnaround of the aircraft by increasing the efficiency of aircraft ground handling operations. The reason was a possible higher throughput of stands, higher profits, more aircraft movements, etc. The research was carried out at Košice Airport. All of the time data included in the research were obtained by measuring individual aircraft ground handling processes directly at Košice Airport. The entire process of changing the layout of ground handling equipment at Košice Airport was consulted with the management of Košice Airport.

The main objective was to determine the distribution of means used, the total number used for the selected type of aircraft, the number of employees participating in equipping one aircraft and the elaboration of precise actions in the order from the arrival of the aircraft to the stand. After compiling the list of tasks, an aircraft was selected and the time of on-block to the time of off-block was measured. Subsequently, the time of individual subtasks were developed with the help of measurements at the selected airport in the usual procedures. After consultations and evaluations, several solutions were proposed to reduce the time of individual tasks. The measurement process was repeated after the application of the changes, and the sets of values were then statistically compared, and a final evaluation was drawn. The essence was increasing the efficiency of operations with the same number of employees as in the usual procedures of the airport. The main changes were focused on the location and preparation of ground handling equipment before landing the aircraft and rolling on the default stand.

### 2.1. Acquisition of Input Data

Measurements of the total turn of the aircraft and partial times of the individual aircraft ground handling operations were taken in the selected months of the summer season. The main reason for choosing the summer season is the frequent repetition of the same types of aircraft with the same configuration characteristics. All contained measurements were performed on one type of aircraft of different airlines at a pre-selected fixed stand. The reason for choosing the same stand for all of the measurements is to ensure comparability of the taxi time, the distance from the arrival terminal, the distances of the ground handling means, the distances of the departure terminals and other factors entering the process.

The first part was measuring the total turnaround time of the aircraft at a pre-selected stand. Subsequently, the research focused on measuring the times of each operation performed during the aircraft ground handling. All standard airport procedures were measured. Following the consultations with the management and employees of Košice Airport, procedural changes in the location of resources and procedures were introduced with the same number of employees. After the changes were introduced, the individual times were measured again and the time savings in the procedures were summarized. Košice Airport is a regional airport with one runway with a length of 3100 m. There is also one taxiway, seven fixed stands for aircraft and one terminal building for both departures and arrivals (Table 1).

**Table 1.** Aircraft ground handling procedures.

| Procedure | Time |
| --- | --- |
| Guide the aircraft to the stand | |
| The wedge of the aircraft | |
| Setting up and connecting a GPU | |
| Apposition of stairs | |
| Exit of passengers | |
| Delivery of fuel vehicles | |
| Refueling JET—A1 | |
| Arrival of aircraft cleaning staff | |
| Cleaning the interior of the aircraft | |
| Delivery of lavatory truck | |
| Dropping of pallets | |
| Delivery of water truck | |
| Filling the aircraft with drinking water | |
| Delivery of a belt conveyor and a tractor with trolleys for checked baggage | |
| Unloading baggage | |
| Baggage loading | |
| Parking of a belt conveyor and a special baggagevehicle | |
| Catering vehicle delivery | |
| Loading catering | |
| Boarding of passengers | |
| Preparation of departure documentation | |
| Transport and inspection of documents stoving + crew | |
| Parking of stairs | |
| Disconnect and park the GPU | |
| Clearance of the aircraft | |
| Rolling out of the stand | |

### 2.2. Number of Employees and Ground Handling Equipment

For the aforementioned procedures, it was necessary to have a good workforce base and a good technical ability of resources. Under ideal conditions, 21 airport employees were needed with ten means of the ground handling equipment (Table 2).

**Table 2.** Employees needed for aircraft ground handling.

| Procedure | Number of Employees |
|---|---|
| Guide the aircraft to the stand and wedge the aircraft (also during the departure procedure) | 1 employee |
| Setting up and connecting a GPU | 1 employee |
| Apposition of stairs | 2 employees |
| Exit of passengers and boarding of passengers | 1 employee |
| Delivery of fuel vehicle + refueling | 1 employee |
| Cleaning the interior of the aircraft | 4 employees |
| Parking of lavatory truck + draining of toilets | 1 employee |
| Delivery of a drinking water vehicle + filling of the vehicle with drinking water | 1 employee |
| Delivery of a belt conveyor and a tractor with trolleys for checked baggage | 1 employee |
| Unloading + loading baggage | 4 employees |
| Vehicle delivery catering + loading catering | 2 employees |
| Preparation of departure documentation | 2 employees |

The ground handling equipment for the aircraft ground handling of the Boeing 737–800:

- 1x Tractor with GPU,
- 2x Mobile stairs for boarding and alighting of passengers,
- 1x Fuel truck with JET-A1,
- 1x Water truck,
- 1x Lavatory truck,
- 1x Special baggage unloading vehicle with built-in belt conveyor,
- 1x Belt conveyor for baggage loading,
- 1x Tractor with trolleys for departure baggage,
- 1x Catering vehicle.

The deployment of the ground handling equipment during ground handling of the aircraft before arrival is shown in the approximate diagram in Figure 1. The location of GHE on arrival was changed to increase the efficiency of the aircraft ground handling, as the positions shown were not effective and valuable time was lost, especially by delivering vehicles to the aircraft. The longest time that was measured in the whole ground handling process was the process of delivery and loading the catering, which came into operation very late in terms of the arrival of the aircraft. The delivery of water and the lavatory truck took more time because they were parked further away in the old airport hangar.

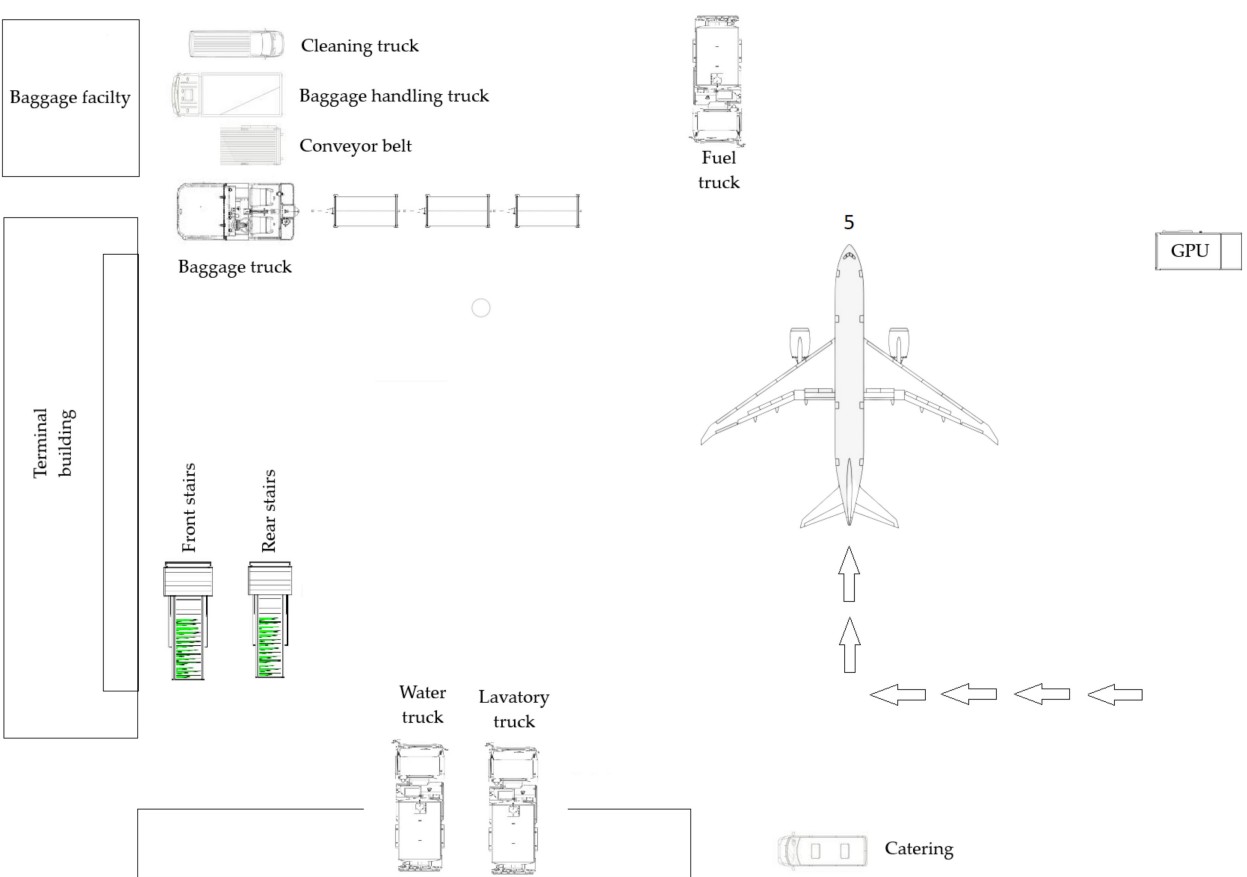

**Figure 1.** Deployment of ground handling equipment before changes were applied.

## 3. Results and Discussion

### 3.1. Initial Measurement Process

Measurements in standard operation were taken during the summer flight schedule on charter flights. In the initial phase, 26 partial processes were measured with 30 partial times of individual aircraft handling operations. The times that we worked with in the research were measured at Košice Airport (Table 3).

**Table 3.** Initial measurements of procedures.

| Procedure | Time |
|---|---|
| Guide the aircraft to the stand | 0:42 |
| The wedge of the aircraft | 0:23 |
| Setting up and connecting a GPU | 1:52 |
| Apposition of stairs | P: 1:25, Z: 1:20 |
| Exit of passengers | 3:56 |
| Delivery of fuel vehicles | 1:42 |
| Refueling JET—A1 | 5:53—4253t |
| Arrival of aircraft cleaning staff | 1:40 |
| Cleaning the interior of the aircraft | 13:15 |
| Delivery of toilet vehicles | 2:30 |
| Dropping of pallets | 1:52 |
| Delivery of drinking water vehicle | 2:41 |
| Filling the aircraft with drinking water | 1:15 |
| Delivery of a belt conveyor and a tractor with trolleys for checked baggage and a special baggage vehicle | V = 2:23, PD = 2:15 |
| Unloading baggage | 9:52 |

**Table 3.** *Cont.*

| Procedure | Time |
|---|---|
| Baggage loading | 10:21 |
| Parking of a belt conveyor and a special baggage vehicle | PD = 0:40, V = 0:48 |
| Catering vehicle delivery | 8:12 |
| Loading catering | 12:45 |
| Boarding of passengers | 12:15 |
| Preparation of departure documentation | 3:35 |
| Transport and inspection of documents stoving + crew | 3:24 |
| Parking of stairs | P: 0:51, Z 1:25 |
| Disconnect and park the GPU | 0:53 |
| Clearance of the aircraft | 0:40 |
| Rolling off the stand + throwing the aircraft engines | 4:25 |

P—front stairs, Z—back stairs, V—special vehicle for baggage, PD—conveyor belt for baggage.

All times listed in the table are average times measured at Košice Airport. All measurements were performed on the same type of aircraft; a Boeing 737–800.

### 3.2. Measurement Process after Changes Were Applied

After the initial times had been measured, the changes were applied. The first aspect that was changed was the position of selecting the means of ground handling of the aircraft.

It should be noted that not every process and every procedure could be subject to the process of increasing the efficiency. Research was focused on increasing the efficiency of the positions of selected ground handling equipment. After the application of the changes, the processes were measured again under identical conditions (Table 4).

**Table 4.** Measurement of procedures after changes were applied.

| Procedure | Time |
|---|---|
| Guide the aircraft to the stand | 0:52 |
| The wedge of the aircraft | 0:36 |
| Setting up and connecting a GPU | 1:15 |
| Apposition of stairs | P: 0:25, Z: 0:58 |
| Exit of passengers | 7:14 |
| Delivery of fuel vehicles | 0:17 |
| Refueling JET—A1 | 8:16—6436t |
| Arrival of aircraft cleaning staff | 0:41 |
| Cleaning the interior of the aircraft | 19:36 |
| Delivery of toilet vehicles | 0:35 |
| Dropping of pallets | 4:46 |
| Delivery of drinking water vehicle | 0:29 |
| Filling the aircraft with drinking water | 1:12 |
| Delivery of a belt conveyor and a tractor with trolleys for checked baggage and a special baggage vehicle | V = 0:28, PD = 0:41 |
| Unloading baggage | 13:48 |
| baggage loading | 21:23 |
| Parking of a belt conveyor and a special baggage vehicle | PD = 0:32, V = 0:41 |
| Catering vehicle delivery | 3:22 |
| Loading catering | 18:36 |
| Boarding of passengers | 15:36 |
| Preparation of departure documentation | 4:18 |
| Transport and inspection of documents stoving + crew | 5:20 |
| Parking of stairs | P: 0:54, Z: 0:52 |
| Disconnect and park the GPU | 1:12 |
| Clearance of the aircraft | 0:46 |
| Rolling off the stand + throwing the aircraft engines | 3:20 |

P—front stairs, Z—back stairs, V—special vehicle for baggage, PD—conveyor belt for baggage.

*3.3. Statistical Evaluation of Measured Data*

      After obtaining all of the necessary input data, research proceeded with the statistical evaluation of the measured times of the aircraft ground handling. The first step was to choose the right method of statistical data evaluation. Each operation from the initial phase was assigned to each operation after certain changes were applied and created the corresponding histogram showing times of each operation. Due to fact that there were two variables in the research, the *t*-test was chosen as a statistical method for the research. Each operation in the aircraft ground handling is interdependent, meaning that if any of the activities are delayed, it will affect the entire process and the total time that the aircraft stays at the airport.

      Table 5 shows operations A1–A5 which were:

- A1—aircraft guidance to stand,
- A2—aircraft wedging,
- A3—GPU delivery and connection,
- A4—approach of the front steps to the aircraft,
- A5—approach of the rear steps to the aircraft.

**Table 5.** Statistical evaluation of measured data A1–A5.

| Operation | N | Mean Value ± Measurement Deviation | Measurement Deviation | Min | Max | P |
|---|---|---|---|---|---|---|
| A1 Before | 78 | 0.490000 ± 0.082664 | 0.087831 | 0.36 | 0.59 | 0.742392 |
| A1 After | 78 | 0.501429 ± 0.076687 | | 0.36 | 0.59 | |
| A2 Before | 78 | 0.322857 ± 0.097248 | 0.069213 | 0.23 | 0.51 | 0.536572 |
| A2 After | 78 | 0.340000 ± 0.088318 | | 0.25 | 0.51 | |
| A3 Before | 78 | 1.330000 ± 0.163401 | 0.204893 | 1.15 | 1.52 | 0.147937 |
| A3 After | 78 | 1.201429 ± 0.073808 | | 1.15 | 1.36 | |
| A4 Before | 78 | 1.244286 ± 0.101136 | 0.073030 | 1.15 | 1.46 | 0.000000 |
| A4 After | 78 | 0.274286 ± 0.092890 | | 0.18 | 0.41 | |
| A5 Before | 78 | 1.211429 ± 0.089709 | 0.150539 | 1.08 | 1.36 | 0.000007 |
| A5 After | 78 | 0.395714 ± 0.099139 | | 0.25 | 0.58 | |

N—number of aircrafts; P—significance level.

      The largest time saving was recorded when performing operation A4 where the time saving at the mean value was 0.97 min. The small-time increase was recorded when performing operation A1, where the required time for operation was increased by 0.011 min.

      Table 6 shows operations A6–A10, which were:

- A6—passenger disembarking,
- A7—delivery of the fuel truck,
- A8—refueling of the aircraft,
- A9—arrival of aircraft cleaning staff,
- A10—aircraft cleaning.

      The largest time saving was recorded when performing operation A7 where the time saving at the mean value was 1.14 min. The small-time increase was recorded when performing operation A6, where the required time for the operation was increased by 0.227 min.

      Table 7 shows operations A11–A15 which were:

- A11—delivery of lavatory truck,
- A12—emptying of the waste tank,
- A13—delivery of the water truck,
- A14—filing the aircraft with drinking water,
- A15—delivery of baggage truck.

**Table 6.** Statistical evaluation of measured data A6–A10.

| Operation | N | Mean Value ± Measurement Deviation | Measurement Deviation | Min | Max | P |
|---|---|---|---|---|---|---|
| A6 Before | 78 | 5.457143 ± 1.422740 | 1.581231 | 3.46 | 7.14 | 0.716990 |
| A6 After | 78 | 5.684286 ± 1.174675 | | 3.46 | 7.14 | |
| A7 Before | 78 | 1.345714 ± 0.149427 | 0.124690 | 1.15 | 1.56 | 0.000000 |
| A7 After | 78 | 0.204286 ± 0.053497 | | 0.10 | 0.26 | |
| A8 Before | 78 | 8.255714 ± 2.235359 | 1.375714 | 5.53 | 11.54 | 0.059329 |
| A8 After | 78 | 6.880000 ± 0.944157 | | 5.20 | 8.16 | |
| A9 Before | 78 | 1.567143 ± 0.480961 | 0.498522 | 1.12 | 2.35 | 0.000653 |
| A9 After | 78 | 0.350000 ± 0.087750 | | 0.26 | 0.51 | |
| A10 Before | 78 | 15.29286 ± 2.839071 | 3.171726 | 12.18 | 19.36 | 0.570825 |
| A10 After | 78 | 16.01143 ± 2.842865 | | 12.18 | 19.36 | |

N—number of aircrafts; P—significance level.

**Table 7.** Statistical evaluation of measured data A11–A15.

| Operation | N | Mean Value ± Measurement Deviation | Measurement Deviation | Min | Max | P |
|---|---|---|---|---|---|---|
| A11 Before | 78 | 2.298571 ± 0.094944 | 0.136835 | 2.16 | 2.46 | 0.000000 |
| A11 After | 78 | 0.375714 ± 0.084628 | | 0.30 | 0.51 | |
| A12 Before | 78 | 2.331429 ± 1.025840 | 1.597868 | 1.36 | 4.46 | 0.680234 |
| A12 After | 78 | 2.592857 ± 1.017738 | | 1.36 | 4.46 | |
| A13 Before | 78 | 2.475714 ± 0.315164 | 0.564383 | 2.26 | 3.16 | 0.000071 |
| A13 After | 78 | 0.420000 ± 0.363822 | | 0.19 | 1.23 | |
| A14 Before | 78 | 0.877143 ± 0.342380 | 0.363816 | 0.48 | 1.18 | 0.976147 |
| A14 After | 78 | 0.872857 ± 0.338561 | | 0.48 | 1.18 | |
| A15 Before | 78 | 2.258571 ± 0.130949 | 0.155349 | 2.10 | 2.46 | 0.000000 |
| A15 After | 78 | 0.288571 ± 0.065683 | | 0.20 | 0.37 | |

N—number of aircrafts; P—significance level.

The largest time saving was recorded when performing operation A15 where the time saving at the mean value was 1.97 min. The smallest time saved was recorded when performing operation A14, where the required time for the operation was reduced by 0.004 min.

Table 8 shows operations A16–A20, which were:

- A16—delivery of baggage trolley and belt conveyor,
- A17—baggage unloading,
- A18—baggage loading,
- A19—parking of belt conveyor,
- A20—parking of special luggage vehicle.

**Table 8.** Statistical evaluation of measured data A16–A20.

| Operation | N | Mean Value ± Measurement Deviation | Measurement Deviation | Min | Max | P |
|---|---|---|---|---|---|---|
| A16 Before | 78 | 2.478571 ± 0.443074 | 0.386695 | 2.06 | 3.15 | 0.000008 |
| A16 After | 78 | 0.398571 ± 0.081328 | | 0.28 | 0.52 | |
| A17 Before | 78 | 12.00143 ± 1.793306 | 1.890841 | 9.36 | 13.48 | 0.697587 |
| A17 After | 78 | 12.29286 ± 1.457083 | | 9.36 | 13.48 | |
| A18 Before | 78 | 16.43000 ± 4.603115 | 4.722167 | 9.56 | 21.23 | 0.578797 |
| A18 After | 78 | 17.47714 ± 3.696849 | | 9.56 | 21.23 | |
| A19 Before | 78 | 0.374286 ± 0.056231 | 0.069727 | 0.30 | 0.46 | 0.366890 |
| A19 After | 78 | 0.348571 ± 0.064402 | | 0.25 | 0.43 | |
| A20 Before | 78 | 0.498571 ± 0.063095 | 0.042857 | 0.42 | 0.59 | 0.042732 |
| A20 After | 78 | 0.455714 ± 0.072309 | | 0.36 | 0.56 | |

N—number of aircrafts; P—significance level.

The largest time saving was recorded when performing operation A16 where the time saving at the mean value was 2.08 min. The small-time increase was recorded when performing operation A17, where the required time for the operation was increased by 0.29 min.

Table 9 shows operations A21–A25, which were:

- A21—delivery of catering truck,
- A22—catering loading,
- A23—passenger boarding,
- A24—preparation of documentation,
- A25—checking the departure documentation.

**Table 9.** Statistical evaluation of measured data A21–A25.

| Operation | N | Mean Value ± Measurement Deviation | Measurement Deviation | Min | Max | P |
|---|---|---|---|---|---|---|
| A21 Before | 78 | 7.850000 ± 1.679305 | 1.595968 | 5.23 | 10.26 | 0.000187 |
| A21 After | 78 | 2.951429 ± 0.460848 | | 2.40 | 3.53 | |
| A22 Before | 78 | 15.76286 ± 2.552076 | 2.596459 | 12.26 | 18.36 | 0.777567 |
| A22 After | 78 | 16.05286 ± 2.204561 | | 12.26 | 18.36 | |
| A23 Before | 78 | 13.56857 ± 1.481288 | 3.380212 | 12.06 | 15.51 | 0.273432 |
| A23 After | 78 | 12.02857 ± 3.185631 | | 6.36 | 15.36 | |
| A24 Before | 78 | 3.957143 ± 0.697823 | 0.757109 | 3.20 | 5.15 | 0.977076 |
| A24 After | 78 | 3.948571 ± 0.706834 | | 3.20 | 5.15 | |
| A25 Before | 78 | 3.748571 ± 0.720079 | 1.138694 | 2.58 | 4.58 | 0.359293 |
| A25 After | 78 | 4.175714 ± 0.972349 | | 2.58 | 5.23 | |

N—number of aircrafts; P—significance level.

The largest time saving was recorded when performing operation A21, where the time saving at the mean value was 4.89 min. The smallest time saved was recorded when performing operation A24, where the required time for the operation was reduced by 0.0085 min.

Table 10 shows operations A26–A30 which were:

- A26—front steps apposition,
- A27—rear steps apposition,
- A28—disconnection and parking of GPU,
- A29—unwedging of the aircraft,
- A30—engine start and roll-off.

**Table 10.** Statistical evaluation of measured data A26–A30.

| Operation | N | Mean Value ± Measurement Deviation | Measurement Deviation | Min | Max | P |
|---|---|---|---|---|---|---|
| A26 Before | 78 | 0.602857 ± 0.311165 | 0.352299 | 0.43 | 1.30 | 0.487083 |
| A26 After | 78 | 0.504286 ± 0.061606 | | 0.43 | 0.58 | |
| A27 Before | 78 | 0.742857 ± 0.409175 | 0.514342 | 0.26 | 1.25 | 0.248252 |
| A27 After | 78 | 0.494286 ± 0.290623 | | 0.24 | 1.12 | |
| A28 Before | 78 | 0.774286 ± 0.306858 | 0.501892 | 0.49 | 1.13 | 0.667253 |
| A28 After | 78 | 0.860000 ± 0.311020 | | 0.49 | 1.13 | |
| A29 Before | 78 | 0.358571 ± 0.099403 | 0.102377 | 0.15 | 0.46 | 0.599740 |
| A29 After | 78 | 0.337143 ± 0.104994 | | 0.15 | 0.46 | |
| A30 Before | 78 | 4.075714 ± 0.986017 | 1.026538 | 2.46 | 5.20 | 0.977457 |
| A30 After | 78 | 4.064286 ± 0.948698 | | 2.59 | 5.20 | |

N—number of aircrafts; P—significance level.

The largest time saving was recorded when performing operation A27 where the time saving at the mean value was 0.24 min. The smallest time saved was recorded when performing operation A30, where the required time for operation was reduced by 0.01 min.

The GHE layout in Figure 2 shows the more efficient positions of the ground handling equipment with the proposed changes. It is this effective deployment on arrival that has reduced the delivery times and the overall time of the aircraft ground handling of the aircraft after arrival and before take-off. All means were delivered in the immediate vicinity of the aircraft, but it was necessary to maintain safe distances to avoid damage to the aircraft and violation of safety procedures.

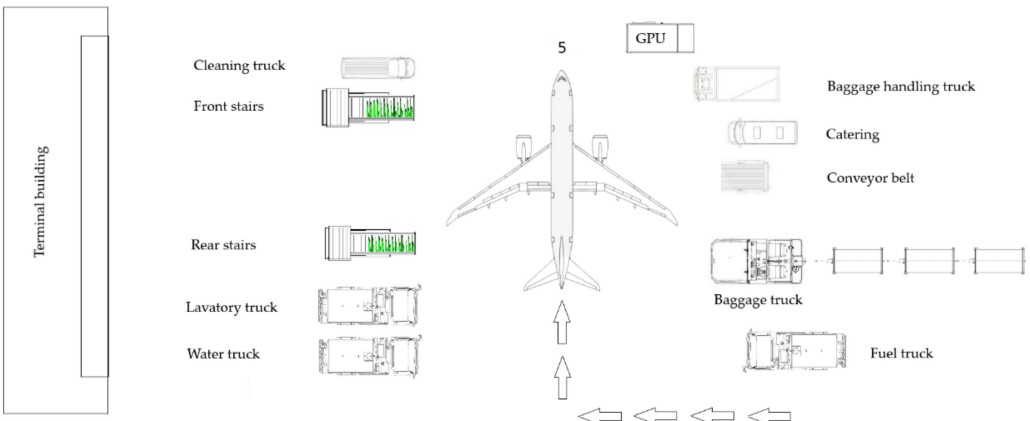

**Figure 2.** Deployment of ground handling equipment after changes were applied.

The statistical evaluation in the research, statistically significant results were obtained in ten cases from 30 statistical tests performed. In these cases, the goal set by the research was confirmed so that the changes in the processes of the aircraft ground handling have had an impact on the individual times of the aircraft ground handling. In nine cases, the target was confirmed at the set level of significance $\alpha = 0.001$ (99.99%) and in one case at the level $\alpha = 0.05$ (95%). Changes in GHE positions as the main goal of increasing the efficiency of aircraft ground handling were made in nine cases, in which a statistically significant result was obtained.

The Table 11 contains the total aircraft ground handling times of selected aircraft in the initial phase before changes were applied and after changes were applied. The results were obtained by summarizing all of the operations of each aircraft in the measurement phases. With 39 aircraft in the initial phase, the total handling time was approximately 910 min (on average, 130 min per group of aircrafts). In the phase after changes were applied, the total handling time for 39 aircraft was approximately 789 min (on average 113 min per group of aircrafts). Subtracting the initial phase time (909.88) from the time after the changes were applied (789.48) equaled 120.40 min, which is approximately a two-hour time saving after applying the changes to increase the efficiency of aircraft ground handling. In one group, a negative value of time was recorded, as the process took approximately 25 min longer after the changes were applied. The times of more efficient operations achieved the expected reduction of time values, but other processes just prolonged the total time. In this case, it is, for example, the passenger disembarking process, which in the case after the changes were applied took longer due to the larger number of passengers on arrival. Another factor was the prolonged cleaning process of the aircraft, for example due to greater pollution of the aircraft, the workload of aircraft cleaning workers and other factors. The process of loading baggage took longer due to the larger number of passengers on departure and other factors that affected the overall time of the aircraft ground handling. Despite this factor, it can be stated that changes in the handling process have led to a reduction in selected partial times.

**Table 11.** Total aircraft ground handling times of selected aircraft groups.

| Measurements | Group 1 | Group 2 | Group 3 | Group 4 | Group 5 | Group 6 | Group 7 | $\sum$ |
|---|---|---|---|---|---|---|---|---|
| Before (t/min) | 108.3 | 107.53 | 148.83 | 136.8 | 133.47 | 140.06 | 134.89 | 909.88 |
| After (t/min) | 80.12 | 132.61 | 117.46 | 109.99 | 119.02 | 112.39 | 117.89 | 789.48 |
| Time saved (t/min) | 28.18 | −25.08 | 31.37 | 26.81 | 14.45 | 27.67 | 17.00 | 120.40 |

*3.4. Discussion*

The main asset of the research was the proposal of process changes in the aircraft ground handling that focused on the position of GHE before the arrival of the aircraft. With simple changes, significant time savings have been achieved, which have led to more efficient process of aircraft ground handling at the selected stand. The applied changes can ensure greater throughput of the stand and the possibility to receive a larger number of aircraft per day, especially in the summer season. The most important finding obtained by this research was that the overall time of aircraft ground handling was reduced by approximately 25 min. These factors will then be reflected in the airport's revenues (airport charges) with various financial factors, such as the departing passenger tax, fuel sales, landing fee and others. Likewise, with a larger number of aircraft, the prestige and airport statistics will increase in the number of checked aircraft per year and in the number of checked passengers per year. The changes that have been introduced, based on this research, do not only affect the processes but also the overall performance and financial indicators of the airport.

From the point of view of further research in this area, it is possible to consider application of same process of increasing the efficiency of aircraft ground handling on all stands at the airport. Due to the scale of measurements, evaluations, and conclusions, it would be easier to think about creating a certain algorithm, program or mobile application that would work with the obtained/entered data, evaluate them, and apply changes to increase the efficiency of aircraft ground handling within the simulation. Another option that seems simpler is to apply similar changes as we introduced in our case to all sites, as the processes are similar. Subsequently, it is necessary to monitor changes and evaluate their effectiveness. A possible change that already has an impact on airport funding is to consider expanding the ground handling base, which would certainly speed up the overall process of delivering the means from the aircraft and to the aircraft during busier days. As there is a significant difference between the summer and the winter seasons in terms of the number of arrivals and departures, the ideal solution would be to rent selected funds, which are in short supply (e.g., mobile stairs, means of filling water and emptying toilets, etc.). Constants (the same number of employees, the same number of aircraft ground handling equipment, the same stand, and the same type of aircraft) were mainly used in the research to obtain the most accurate and homogeneous results. As research continues, it is possible to focus on multiple inputs of a diverse nature and create research to develop a complex program or application that would work with these values, evaluate them, and then simulate them into outputs.

**4. Conclusions**

The goal of the research was to increase the efficiency of aircraft ground handling at the selected airport. In the changes in the deployment of selected ground handling equipment, the individual processes were shortened, as well as the total stay of the aircraft at the airport after landing and before take-off. From the results of the research, it can be stated that the goal of the research was achieved.

After measuring the times of individual processes in the initial phase of research and monitoring the possible application of the suitable changes, the changes were introduced into the aircraft ground handling processes of the aircraft and the measurements were repeated under the same conditions. Both measurement processes (before and after

changes were applied) provided a sample of measured times (2340), which was statistically evaluated in the research. In the process of statistical evaluation, descriptive statistics were used, which is a method for creating an overview of the obtained data. Subsequently, the research used inductive statistics, which is used to draw conclusions from the measured values by using the *t*-test. The statistical results were used to verify whether the changes introduced into the processes were statistically significant. Due to the complexity of the aircraft ground handling processes, statistical verification was performed on all processes and not only on processes in which changes were applied.

Overall, it can be stated that in the research and measurement of partial times of specific aircraft ground handling operations, several operations were found where it is possible to achieve significant time savings. The most significant time saving was achieved with the delivery of the catering truck by 4.89 min, mainly due to the large distance of the original position of the catering truck from the stand. Research has also confirmed that certain operations cannot be done faster. An example is guiding the aircraft to a stand where there was no time saved. Thus, it can be concluded that changes made in the ground handling of an aircraft led to a significant reduction in the turnaround time, which was shorter by approximately 25 min.

**Author Contributions:** Conceptualization, S.S., P.K., M.P., S.M. and Ľ.K.; data curation, S.M. and M.Č.; methodology, S.S., P.K. and Ľ.K.; formal analysis, S.S., P.K., M.P. and S.M.; validation S.M. and M.Č.; supervision, S.S. and P.K.; resources, M.P. and S.M.; writing—original draft preparation, S.M. and M.Č.; writing—review and editing, S.S., P.K., M.P. and Ľ.K. All authors have read and agreed to the published version of the manuscript.

**Funding:** This article was supported by project New possibilities and Approaches of Optimization within Logistic Processes in the field of Transport and Transport Systems, ITMS code: 313011T567.

**Institutional Review Board Statement:** Not applicable.

**Informed Consent Statement:** Not applicable.

**Data Availability Statement:** Not applicable.

**Conflicts of Interest:** The authors declare no conflict of interest.

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
