# Peer review of "Increasing the Efficiency of Aircraft Ground Handling—A Case Study"

_aerospace, doi:10.3390/aerospace9010002_

Round 1

Reviewer 1 Report

The aim of this paper is to fasten ground handling procedures by changing ground equipment positions. Despite the authors state more than once that the methodology is based on optimization, as far as I understand no optimization technique is used. In fact, no mathematical model is shown. The methodology is quite mysterious, and no detailed information is provided on many aspects. For example, how did you select the changes pre and post “optimization”? Is there any mathematical model? In the paper, only “measurements” are shown, without detailing from where values come from.

Moreover, the literature of aircraft ground handling is wide and the topic is not new. So which is the contribution of the work? The paper shown results only for an airport, without any general consideration. To me, it seems more a case study. Then you should revise the title, which is very general. Moreover, nothing is said regarding the airport you analyse. Given that the main variable on which you work is ground equipment location, then it should be helpful if you provide some information on it (distances, airport layout…).

For these main reasons, I’m sorry but I think that the article is not appropriate for publication, and my opinion is to reject it. In the following you can find some observations which would help improving your paper, for future publication.

Results are not correctly elaborated and described. In Table 2, is “time” an average? If yes, which is the standard deviation? Moreover, is it averaged on similar aircraft? Some ground procedures depend on the aircraft type, did you consider it?

Moreover, in the results section, you are describing what already shown in tables. Tables are enough, please be more concise when describing them. Moreover, I don’t agree with some sentences. For example, for A1, I would not say that a time equal 0.50 should be considered “higher” than 0.49…

The paper il well-written, but the English should be improved and many errors can be found. Moreover, many sentences are not clear. For example:

  • Line 33: “the research”? which one? If the one of the paper, then use “this”. Otherwise specify research on what.
  • Line 46: the length of the handling is a direct factor of what? Of the time required for the aircraft ground handling? I don’t understand
  • Consists of procedures which consists of operations…. Sounds repetitive.
  • Line 70: check English.
  • Line 70: which problem?

Line 50: also the number of passengers is a factor (for the disembarking process duration), and not only the amount of luggage

Line 100: Which consultations? The methodology should be better described.

Lines 103: this has been already said more than once… also at line 138, again. I suggest being less repetive.

Author Response

Dear reviewer,

Authors

Reviewer 2 Report

This is a nice piece of academic research and I think it is well worth publishing. The analysis is straightforward, which is nice, and I have only two minor concerns: First, language might be improved a bit, but this is not a big thing. Of greater concern are the Tables 5 to 10: I understand what the authors want to say, but from the presentation (in the text as well as the tables) it is unclear, what is the actual p-value and what is the critical p-value for alpha<0.05? I don't understand it in the tables and also not in the text. Furthermore "," needs to replaced by "." in the numbers. 

Author Response

Dear reviewer,

Authors

Round 2

Reviewer 1 Report

In the revised version of the paper some improvements have been made (for example providing additional details regarding the case study). However, I think that some other major revisions are needed.

  • My main concern regards the use of the term “optimization”. In scientific terms, “optimization” refers to some specific mathematical models and algorithms. How did you perform “optimization”? Which are the objective function and constraints? Please, clarify.
  • I think that the results section should be strongly revised (pages 7-12). This observation was made also in the first revision round but, in my opinion, the authors did not address it properly. Specifically, tables’ description is lengthy and not crispy. I think it should be made clearer and more focused. There is no need to describe in detail every number of the tables, they are already shown in the tables. You should however focus on what results show: what interesting observation can be drawn from results? Maybe, it should be more useful to focus on some specific processes, for example the ones for which the time savings are higher. Otherwise, the reader gets lost and does not understand the point of your analysis.
  • In the discussion section you report some results, and very little discussion is made. I suggest moving results in previous section - which maybe can be re-named “results and discussion” – and enlarge the discussion. Then you should think an appropriate conclusion section, in which you highlight the contribution of your work and of the obtained results.
  • Title: Optimization of aircraft ground handling – A case study

Author Response

Dear reviewer,

Authors

Round 3

Reviewer 1 Report

Thank you for considering my observations, I think that the paper has improved